# The Dichotomous Role of Inflammation in the CNS: A Mitochondrial Point of View

**DOI:** 10.3390/biom10101437

**Published:** 2020-10-13

**Authors:** Bianca Vezzani, Marianna Carinci, Simone Patergnani, Matteo P. Pasquin, Annunziata Guarino, Nimra Aziz, Paolo Pinton, Michele Simonato, Carlotta Giorgi

**Affiliations:** 1Department of Medical Sciences, University of Ferrara, 44121 Ferrara, Italy; vzzbnc@unife.it (B.V.); crnmnn@unife.it (M.C.); simone.patergnani@unife.it (S.P.); psqmtp@unife.it (M.P.P.); paolo.pinton@unife.it (P.P.); 2Laboratory of Technologies for Advanced Therapy (LTTA), Technopole of Ferrara, 44121 Ferrara, Italy; grnnnz@unife.it (A.G.); nimra.aziz@unife.it (N.A.); michele.simonato@unife.it (M.S.); 3Department of BioMedical and Specialist Surgical Sciences, University of Ferrara, 44121 Ferrara, Italy; 4Maria Cecilia Hospital, GVM Care & Research, 48033 Cotignola (RA), Italy; 5School of Medicine, University Vita-Salute San Raffaele, 20132 Milan, Italy

**Keywords:** neuroinflammation, mitochondria, neurodegeneration, multiple sclerosis, Parkinson’s disease, Alzheimer’s disease, ischemic stroke, epilepsy

## Abstract

Innate immune response is one of our primary defenses against pathogens infection, although, if dysregulated, it represents the leading cause of chronic tissue inflammation. This dualism is even more present in the central nervous system, where neuroinflammation is both important for the activation of reparatory mechanisms and, at the same time, leads to the release of detrimental factors that induce neurons loss. Key players in modulating the neuroinflammatory response are mitochondria. Indeed, they are responsible for a variety of cell mechanisms that control tissue homeostasis, such as autophagy, apoptosis, energy production, and also inflammation. Accordingly, it is widely recognized that mitochondria exert a pivotal role in the development of neurodegenerative diseases, such as multiple sclerosis, Parkinson’s and Alzheimer’s diseases, as well as in acute brain damage, such in ischemic stroke and epileptic seizures. In this review, we will describe the role of mitochondria molecular signaling in regulating neuroinflammation in central nervous system (CNS) diseases, by focusing on pattern recognition receptors (PRRs) signaling, reactive oxygen species (ROS) production, and mitophagy, giving a hint on the possible therapeutic approaches targeting mitochondrial pathways involved in inflammation.

## 1. Introduction: The Cellular Players of Neuroinflammation

The new century, together with technological innovations, brought new insight into the intrinsic communication between the central nervous system (CNS) and the innate immune response. It was a common thought that the brain was a privileged tissue of our body, due to the presence of the blood–brain barrier (BBB) that would have avoided the access of immune cells [1,2]. This hypothesis has been challenged by an increasing number of studies, becoming nowadays an obsolete consideration, even though the CNS still conserves some unique immunological features [3]. Specifically, immune cells reside at the meninges granting surveillance to the brain, and meninges are provided of lymphatic vessels, able to drain large particles and immunomodulatory cytokines directly to the peripheral immune system through lymph nodes connections [4,5]. Nevertheless, the resident key players of the neuroimmune system are glial cells. These CNS immune cells are classified as macroglia (oligodendrocytes and astrocytes) and microglia, they regulate several physiological processes required for neuronal survival and brain function. As far as we are now aware, besides being part of glial cells, oligodendrocytes do not have a major role in the physiological neuroinflammation, since they mainly provide physical and metabolic support to neurons promoting the myelinating process [6]. Noticeably, oligodendrocyte gap junctions’ deficiency due to genetic defects has been associated with increased neuroinflammation in mouse models, indicating that the altered expression of connexins in oligodendrocytes, besides being a consequence of inflammation, can also promote a proinflammatory environment [7]. Astrocytes are the most numerous glial cells of the CNS, exerting diverse roles, such as the regulation of synaptic plasticity and, more broadly, the control of brain homeostasis, also by coordinating local energy metabolism. Importantly, they also play a role in neuroprotection by maintaining the BBB intact, due to their tight interactions with the cerebrovascular endothelium [8,9]. Furthermore, astrocytes release proinflammatory cytokines, such as tumor necrosis factor α (TNF-α), which besides boosting the local inflammatory response by acting on microglia and neurons, is important in facilitating lymphocytes crossing the BBB into CNS parenchyma [10]. Accordingly, abnormal astrocytes activation, mainly characterized by hypertrophy of soma and processes, plays a key role in the neuroinflammatory process, also owed to the strict communication with microglia [11]. Being firstly described a century ago by Pio del Rio Hortega as the ‘third element’ of the CNS [12], microglia cells are now defined as the innate immune cells of the CNS characterized by the expression of CX3CR1, CD11b, Iba1, and F4/80 markers, by their myeloid origin, and by their phagocytic ability [13]. Microglia exert different functions in the CNS: they are responsible for sensing changes in the surrounding microenvironment, including both physiological changes and pathogens invasion, thus activating either their housekeeping or defense function [14].

Neuroinflammation is a natural process of defense, precisely and timely regulated, which includes a proinflammatory phase aimed to neutralize the danger, and an anti-inflammatory phase that restores the tissue homeostasis by activating the regenerative processes. While an acute neuroinflammatory response reduces injury by contributing to the repair of damaged tissue, chronic glial activation, which results from persistent stimuli, is a fundamental component of neurodegenerative diseases, and contributes to neuronal dysfunction, and therefore to CNS diseases progression [15]. As a consequence, the neuroimmune response performed by activated glial cells has a dichotomous role in the CNS. On one side, it induces the activation of repairing and regenerating mechanisms (i.e., remyelination), while on the other, the uncontrolled release of inflammatory mediators as proinflammatory cytokines, reactive oxygen species (ROS), and nitric oxide (NO) boost a chronic neuroinflammatory state, and is potentially dangerous for the neighboring cells. The aberrant release of these inflammatory molecules, together with the consequent upregulation of immune receptors on the other CNS cells, lead to tissue damage and the consequent activation of peripheral B- and T-cell responses due to the meningeal lymphatic system drainage [16]. This cascade of events enhances the inflammatory process owing to the synergistic action of microglia and lymphocytes against the antigen presenting cells [17,18]. Acute neuroinflammation usually takes place during infectious disease or during chronic autoimmune disorders such as multiple sclerosis (MS), but recent evidence has suggested how prolonged neuroinflammation is a ubiquitous pathological sign of several neurodegenerative diseases such as Parkinson’s and Alzheimer’s diseases (PD and AD) [19,20,21]. Accordingly, the close link between neuroinflammatory state and neurodegeneration suggests that neuroimmune mechanisms might trigger neuronal degeneration, resulting in neurotoxicity and neuronal cell loss [22,23]. Interestingly, the presence of mitochondrial dysfunctions both in neurodegenerative and neuroinflammatory CNS pathologies might represent the key connection between chronic immune activation and neuronal degeneration [24,25,26]. Mitochondria are organelles of endosymbiotic bacterial origin involved in various cellular functions, from the regulation of energy production and metabolism to the control of cell proliferation and programmed cell death [27]. Noticeably, mitochondria are also endowed with the ability to sense and react to cellular damage and to promote efficient host immune response by producing secondary messengers fundamental in the activation of immune cells and by contributing to the activation of inflammasomes, i.e., of the intracellular protein complexes that detect and respond to danger stimuli. Therefore, it is not surprising that increasing literature is supporting the central role of these organelles in the pathogenesis of both inflammatory and neurodegenerative CNS disorders.

In this review we are going to discuss the central role of mitochondria in driving and maintaining the neuroinflammatory process present either in chronic primarily inflammatory CNS diseases such as MS, chronic non-inflammatory neurodegenerative diseases such as PD and AD, and also in non-primarily inflammatory CNS pathologies such as epilepsy and ischemic stroke. We are going to focus on the mitochondrial pathways regulating inflammation in microglia and astrocytes because, to the best of our knowledge, these two cell types are the most involved in triggering and sustaining the neuroinflammatory process. Finally, we are going to discuss the current therapies aimed to reduce neuroinflammation in the cited pathologies.

## 2. Role of Mitochondria in Neuroinflammation

Neuroinflammation is an innate inflammatory response within the CNS against harmful and toxic stimuli, mediated by the activation of resident immune cells, by the recruitment of peripheral lymphocytes and, lastly but most importantly, by the production of cytokines, chemokines, ROS, and other proinflammatory secondary messengers. The main cellular players involved in the neuroinflammatory process are glial cells, such as astrocytes and microglia. For a long time, glial cells residing in a healthy brain were defined as inactive. Following damage or infection, glial cells become “activated”, even though the terms “resting” and “activated” are vague and obsolete due to the high plasticity of these cells, which have shown to be able to dynamically shift between a spectrum of different phenotypes [28] (Figure 1). In fact, the advent of in vivo techniques, such as 2-photon microscopy, allowed the discovery that in their “resting” state, microglial cells are instead highly active, by surveying their microenvironment with extremely motile processes and protrusions [29]. Additionally, astrocytes, the most abundant glial cell population, participate in the immune and inflammatory responses of the CNS by sensing both exogenous and endogenous material through the expression of specific receptors. Indeed, even if mainly expressed by microglial cells, pattern recognition receptors (PRRs) that are fundamental for the primary recognition of infectious agents and of endogenous danger signals, are also expressed by astrocytes [30]. PRRs, localized on the cell surface, in the endosomes and also in the cytoplasm, upon the recognition of a specific antigen lead to intracellular signaling cascade ending with the release of proinflammatory mediators [31]. It is important to underline that astrocytes mainly rely on microglia for their activation. In fact, microglia control the surrounding microenvironment by using their dynamic ramifications to sense and detect any occurring alteration in brain homeostasis: once in contact with dangerous molecular factors, microglia acquire a less ramified phenotype, starting their immunomodulatory activity either by phagocytosis or by proinflammatory factors secretion [32]. Several molecular pathways are involved in activating and maintaining the inflammatory state within the CNS: PRRs signaling, cytokine receptor signaling, triggering receptor expressed on myeloid cells-2 (TREM2) signaling, ROS production, and secretion [32,33]. Interestingly, an increasing number of studies demonstrate the direct involvement of mitochondria in the modulation of the innate immune response by their participation in PRRs signaling, ROS production, and thus inflammasome assembly [34,35,36], as shown in Figure 1. Particularly, mitochondrial damage and/or dysfunction such as mitochondrial depolarization or excessive ROS production promote a selective autophagic process called mitophagy. Due to the importance of mitochondria in regulating neuroinflammation, mitophagy represents a key factor in modulating damage-associated molecular patterns (DAMPs) response, by preventing their release both in the cytoplasm and in the extracellular space. Therefore, its alteration has a fundamental role in the establishment of a proinflammatory environment in the development of CNS disorders. We are going to focus our attention on these three latter mechanisms, PRRs signaling, ROS production, and mitophagy, briefly describing their involvement in the neuroinflammatory process and then describing their participation in MS, PD, AD, ischemic stroke, and epilepsy.

### 2.1. PRRs Signaling: Focus on cGAS-STING Pathway

As mentioned above, the innate immune system is able to recognize pathogens through the presence of the receptor families of PRRs [37]. These receptors are present on inflammatory cells like macrophages, neutrophils, dendritic cells, microglia, and astrocytes. PRRs can be on the cell membrane, such as the Toll-like receptors (TLR), or can be present in the intracellular compartments as for the nucleotide-binding oligomerization domain-like receptors (NLRs) and absent in melanoma 2 (AIM2)-like receptors (ALRs) [38]. Physiologically, PRRs play a protective role in host defense against damaging signals, but their abnormal activation leads to chronic inflammation. As a part of the innate immune system, inflammation is initiated when PRRs detect pathogen-associated molecular patterns (PAMPs), such as microbial nucleic acids, lipoproteins, and carbohydrates. On the other hand, in 2002 Matzinger developed the “danger theory”, stating that the human body, in absence of infection, uses the same system to signal tissue damage (sterile inflammation), and activate repairing mechanisms [39]. In fact, PRRs are also able to detect DAMPs that are commonly released from injured cells following stress conditions [40]. Due to their bacterial origin [41], mitochondria represent an important source of DAMPs, thus playing an important role in immune system activation and induction of sterile inflammation. Under stress conditions, the outer mitochondrial membrane (OMM), can be damaged triggering the subsequent disruption of the inner mitochondrial membrane and the release of mitochondrial components, such as mitochondrial DNA (mtDNA), N-formylated proteins, and cardiolipin in the cytoplasm [42,43]. The release of mitochondrial components triggers the activation of different PRRs, such as inflammasomes, cyclic GMP-AMP synthase (cGAS), and TLRs [44]. Notably, the expression of most PRRs in the CNS is not restricted to microglia and astrocytes, but it also occurs in neurons indicating that they not only suffer neuroinflammation, but they might be involved in its regulation [45,46,47].

mtDNA is a circular molecule of double-stranded (ds)DNA enriched in bacterial hypomethylated CpG island therefore highly capable of eliciting the PRRs response by binding to the TLR9 [48]. Recently it has been identified, through a strategy that combined quantitative mass spectrometry with conventional protein purification, a novel sensor of cytosolic dsDNA able to trigger the type-I interferon (IFN) pathway: the cGAS [49]. Briefly, by binding to two molecules of cytosolic dsDNA, cGAS converts adenosine triphosphate (ATP) and guanosine triphosphate (GTP) into the second messenger 2′,3′-cyclic GMP-AMP (cGAMP) [50], which binds and activates the ER-resident protein stimulator of interferon genes (STING) [51]. This bond causes conformational reorganization of STING, which allows its phosphorylation by the TANK-binding kinase 1 (TBK1) in the endoplasmic reticulum (ER)-Golgi intermediate compartment. After its activation, STING phosphorylates the interferon regulatory factor 3 (IRF3) which dimerizes, translocates to the nucleus and induces expression of type I IFNs. Moreover, STING activates the IĸB kinase complex, which phosphorylates IĸB, an inhibitor of nuclear factor-ĸB (NF-ĸB). IĸB degradation allows the translocation of NF-ĸB into the nucleus and the consequent induction of inflammatory cytokines [52,53]. This pathway is physiologically activated during pathogen infections, and its activation is important for the correct pathogen response. However, in case of sustained dsDNA presence in the cytoplasm, its continued activation might result in abnormal neuroinflammation. Interestingly, mitochondria disruption represents a great source of cytoplasmic dsDNA. Accordingly, mitochondrial damage is not only a cause, but is also a consequence of neuroinflammation, resulting in the release of mtDNA in the cytosol and also in the extracellular space. It is therefore not surprising how the cGAS-STING pathway has recently assumed considerable importance for the understanding the molecular bases beyond various neuroinflammatory and neurodegenerative diseases (NDDs) [54].

### 2.2. PRRs Signaling: Focus on NLRP3 Inflammasome

Another important family of PRRs responsible for the early recognition of PAMPs and DAMPs expressed by microglia and astrocytes are the NLRs. The NLR family is characterized by the presence of a central nucleotide and oligomerization domain (NACHT), which is common to all members of the NLR family, flanked by C-terminal leucine-rich repeats (LRR) and N-terminal caspase (CARD) or pyrine (PYD) recruitment domains. LRR regions are responsible for ligand detection, while the CARD and PYD domains mediate the protein–protein interactions for the activation of downstream signaling [55]. Noticeably, some NLRs, once activated following the detection of PAMPs or DAMPs, can lead to the formation of a multiprotein complex called “inflammasome” [56]. Outstandingly, besides the inflammasomes derived from the NLR family, such as NLRP1, NLRP3, and NLRC4, also non-NLR proteins such as ALRs and pyrin can lead to inflammasomes assembly [57]. Among the cited ones, the NLRP3 (nucleotide-binding domain and leucine-rich repeat containing protein 3, also known as NALP3) inflammasome is the most studied and it is present both in microglia and astrocytes, even though a later study reported that NLRP3 was predominantly active in microglia [58,59]. The NLR domain represents the sensory component of the inflammasome, which binds to the amino-terminal domain of the adaptor apoptosis-associated speck-like protein containing CARD (ASC), once dangerous molecules are detected. ASC forms a bridge with the CARD domain, which contains the pro caspase-1, which subsequently self-catalyzes to its active form caspase-1 leading to the production of the proinflammatory cytokines IL-1β and IL-18 [60]. Moreover, caspase-1 is accountable for the cleavage of gasdermin D, which promotes inflammasome-associated pyroptotic cell death by producing pores in the cell membrane allowing also IL-1 β and IL-18 secretion [60]. Notably, in the resting state, NLRP3 is localized in the cytosol and upon activation it relocates in mitochondria and at the mitochondria associated membranes (MAMs) together with its partner ASC [61]. The triggering signals for inflammasome assembly and delocalization are a variety of exogenous and endogenous stimuli such as microbial infections, extracellular ATP, ROS, and mtDNA. As for the cGAS-STING pathway, also the NLRP3 inflammasomes, and more broadly all the inflammasomes, have a protective role against pathogen infections and also against metabolic toxic waste accumulation, by sustaining the innate immune response in order to defeat the harmful stimuli. On the other hand, sustained NLRP3 activation due to abnormal amounts of misfolded protein or metabolic by-products, leads to chronic neuroinflammation, an ideal environment for the development of CNS disorders [62]. In this latter context, the ability of mitochondria components and products to activate the NLRP3 inflammasome indicates that it is responsible for sensing mitochondrial dysfunction, thus explaining the frequent association of mitochondrial damage with inflammatory diseases [61]. In particular, different studies reported that microglial NLRP3 inflammasome activation is a key contributor to the development of the neuroinflammatory process during neurodegeneration. Indeed, microglial NLRP3 activation has been shown to be triggered by pathogenic protein aggregates such as β-amyloid protein (Aβ) and α-synuclein (α-Syn), related to the development of amyotrophic lateral sclerosis (ALS), AD and PD [63], but also by the neurotoxin 1-methyl-4-phenyl-1,2,3,6-tetrahydropyridine (MPTP) commonly used to model PD in mice [64]. Moreover, it has been reported that the microglial activation of the NLRP3 inflammasome drives tau pathology in a mouse model of frontotemporal dementia, shedding a light on the role of microglia in the development of AD [65]. Lastly, also ceramide, the sphingosine-based lipid-signaling molecule linked to the development of numerous pathophysiological processes in the CNS including AD, has been reported as a modulator of NLRP3 inflammasome assembly [66]. Interestingly, STING has been proposed to activate the NLRP3 inflammasome at least by two distinct mechanisms upon cytosolic DNA stimulation, indicating a connection between cGAS-STING and NLRP3 pathways. On one hand STING interacts and recruits NLRP3 allowing its localization in the ER and facilitating the inflammasome formation, on the other hand the interaction reduces K48- and K63-linked polyubiquitination of NLRP3 favoring the inflammasome activation [67]. In conclusion, besides being fundamental in the physiological process of pathogen-driven immune response, abnormal NLRP3 inflammasome activation exerts a fundamental role in the progression of neuroinflammation and, consequently, in the development of a variety of neurodegenerative diseases, representing a promising target for therapies.

### 2.3. Reactive Oxygen Species

For several years now, it has been widely known that cytokine-activated microglia produce ROS and that, as stated before, ROS are responsible for activating microglia [68]. Accordingly, oxidative damage is both a cause and a result of the neuroinflammatory process leading to the neurotoxic effects observed in different NDDs. Reactive species, also called free radicals, include reactive oxygen and nitrogen species. ROS are a physiological by-product of oxygen metabolism and exert significant roles in cell signaling. They are mainly generated by mitochondria and include oxygen radicals such as superoxide (O_2_^•−^), hydroxyl (•OH), peroxyl (RO_2_^•^), and alkoxyl (RO^−^), and also non-radical oxidizing agents easily convertible into radicals, such as hypochlorous acid (HOCl), ozone (O_3_), singlet oxygen, and hydrogen peroxide (H_2_O_2_) [69]. In addition, reactive nitrogen species (RNS), such as nitric oxide (NO), are produced at low levels during the mitochondrial oxidative phosphorylation (OXPHOS) [70]. When the redox state is balanced, ROS act as second messengers in different signaling pathways, contributing to the conservation of cellular functionality [71]. However, when oxygen homeostasis is not maintained, the redox balance is compromised thus leading to ROS accumulation, with the consequent assembly of NLRP3 inflammasome and the disruption of the OMM, which, in the CNS, results in the induction of the neuroinflammatory state. The exacerbated production of ROS leads to the activation of glial cells resulting in proinflammatory cytokines release, which in turn stimulates the apoptosis of pericytes, important regulators of the BBB, via ROS augmentation [72]. Furthermore, in damaged mitochondria, the rate of O_2_^•−^ formation is increased by the loss of electrons, leading to the formation of H_2_O_2_. O_2_^•−^ can therefore react with NO, which is produced by cellular NO synthase, with the consequent formation of peroxynitrite, leading to increased cell damage [73,74]. Since the brain is one of the highest ATP-demanding organs, OXPHOS is highly active in CNS cells, and is responsible for the maintenance of neuronal function like synaptic transmission and preservation of neuronal potential [75,76,77]. Therefore, prolonged mitochondrial dysfunction leading to the failure of ATP production and to increased ROS and RNS production is considered at the base of neuronal cell loss in neuroinflammation [76,78,79,80]. Nevertheless, it is important to recall the fundamental role of ROS in the maintenance of tissue homeostasis when redox balance is preserved, implying that too aggressive antioxidant therapies might compromise also the physiological role of ROS and thereby, CNS functions. Overall, a comprehensive understanding of the fine redox tuning and ROS production in neuroinflammation and during NDDs progression may help to develop new, antioxidant-based adjuvant therapies.

### 2.4. Mitophagy

Mitochondria, as all the other cellular organelles, experience a continuous turnover through the coordinated degradation, recycling, and new synthesis of their constituent elements [81]. In the CNS, neuronal cell functions highly depend on the efficiency of mitochondria, either in their ability to produce energy and balance inflammatory response or in their capacity of undergoing selective degradation. This latter function is known as mitophagy, a physiological process aimed to specifically eliminate damaged mitochondria, or to remove all mitochondria in specific developmental phases, in order to preserve tissue homeostasis [82]. During the neuroinflammatory process, a lot of mitochondria by-products are generated inducing mtDNA mutations and the alteration of the mitochondrial membrane potential (Ψm), exacerbating the inflammatory state [83]. In this scenario, mitophagy plays a pivotal role, by removing damaged mitochondria thus reducing the cellular damage and avoiding neuronal cell loss, preserving CNS function. This mechanism is based on ubiquitin-dependent and receptor-dependent signals released from damaged mitochondria [84].

The best-characterized mitophagic pathway in mammalian cells is the PTEN-induced putative kinase 1 (PINK1)/Parkin pathway [85], which relies on ubiquitin mediated degradation. This process is triggered by a decrease of the Ψm, due to mitochondrial permeabilization, which leads to the recruitment of PINK1 at the OMM. At this point, PINK1 enrolls the E3 ubiquitin ligase Parkin, leading to the ubiquitination of different OMM mitochondrial proteins [86]. Polyubiquitinated mitochondrial proteins are then associated with the ubiquitin-binding domains of autophagy receptors inducing the formation of the autophagosome that will be subsequently degraded by its fusion with the lysosome [87]. At least five specific ubiquitin-binding autophagy receptors were identified to connect ubiquitinated mitochondria to the phagosomes. However, it remains to be clarified if one among p62/sequestosome 1 (p62/SQSTM1), nuclear dot protein 52 (NDP52), neighbor of Brca1 (NBR1), tax 1 binding protein 1 (TAX1BP1), and optineurin (OPTN) is effectively essential for mitophagy [88,89].

The other pathway that regulates mitophagy is dependent on proteins localized on the OMM that act as receptors, such as B-cell lymphoma 2 nineteen kilodalton interacting protein 3 (BNIP3), Nix, Bcl-2-like protein 13, and FUN1. These proteins all contain the LC3-interacting region (LIR) motif, which is responsible for the recruitment of the autophagosomal machinery by the direct interaction of the mitochondria with LC3/GABARAP family members [81]. Being strictly correlated with the inflammatory process, due to its scavenger activity, the mitophagic pathway is commonly altered in CNS disorder, representing an appealing target for therapies [82].

## 3. From Chronic Neuroinflammation to Neurodegeneration: Multiple Sclerosis, Parkinson’s, and Alzheimer’s Disease

All neurodegenerative disease, including MS, PD, and AD share a common feature: chronic aberrant inflammation (Figure 2). This condition starts with a systemic inflammation that activates the immune response in the CNS, particularly throughout the priming of brain resident microglia. This leads to the subsequent release of inflammatory mediators and the consequent upregulation of the immune response. As described before, mitochondria take part in this unfavorable condition. Particularly, activated microglia increase the production of mitochondrial oxidative species, such as ROS and RNS, which can oxidize and damage lipids, nucleic acids, proteins, and polysaccharides leading to further mitochondrial damage. All these hostile conditions act as a feedback sufficient to sustain a stressful condition that promotes tissue damage and chronic inflammation, leading to nervous tissue degeneration. Accordingly, several studies have demonstrated that inflammation and a perturbed mitochondrial population exacerbate the outcome of neurodegenerative diseases.

### 3.1. Multiple Sclerosis

MS is the commonest primary demyelinating disease of the brain. MS displays a great inter-individual variability in disease course and severity [90]. About 1–3% of the affected people have a benign form of MS, in which any severe disability occurs after several years [91]. About 10–15% of MS patients present a progressive primary form, where symptoms and disabilities gradually get worse over time. Lastly, the majority (80–85%) present a relapsing-remitting form, where an attack is followed by a time of recovery with few or no symptoms, called remission.

MS is a T-cell–mediated autoimmune disease characterized by demyelination, gliosis, and neuronal cell loss [92]. The association of MS with a strong inflammatory process has been challenged over the years, but it is now evident that cortical demyelination occurs in association with neuroinflammation [93]. Inflammatory pathogenic T cells enter the CNS to initiate the immunological cascade leading to the activation of residing microglia and astrocytes, which, together with the further participation of B cells and dendritic cells, finally trigger the chronic CNS inflammation [94]. Indeed, increased levels of proinflammatory cytokines like IFN-γ, IL-2, IL-18, and TNF-α, have been found in human samples obtained from MS patients [95,96]. To become biologically active TNF-α has to be cleaved by a disintegrin and metalloproteinase (ADAM-17) called TNF-α-converting enzyme (TACE). Interestingly, elevated levels of TNF-α and TACE mRNA were found in peripheral blood mononuclear cells (PBMCs) of MS patients, without an ex vivo stimulation [96]. During their migration into CNS, activated T cells express matrix metalloproteinases (MPPs) that drive the lysis of the dense subendothelial basal lamina, resulting in progressive tissue damage. Remarkably, increased levels of MMPs were observed in cerebrospinal fluid (CSF) of MS patients [97]. Notably, MPPs not only mediate tissue damage, but also regulate the inflammatory reaction through TNF-α processing [98]. Consistent with this, the most frequently used treatment for MS, namely IFN-β, acts by reducing MPPs expression and therefore by interfering with the passage of activated T cells into CNS [97].

The neuroinflammatory process can be also triggered by pathogen infection that causes the release of proinflammatory mediators within the CNS. It is therefore not surprising that different pathogens such as *Mycoplasma pneumoniae*, *Staphylococcus aureus*, *Chlamydia pneumoniae*, Epstein Barr, and Herpes viruses are associated to the development or exacerbation of MS, due to their ability to affect the cGAS-STING pathway [99]. Interestingly, it has recently been shown that the antiviral drug ganciclovir inhibits the proliferation of microglia in experimental autoimmune encephalomyelitis (EAE), the most commonly used experimental model for the human inflammatory demyelinating disease, by modulating the cGAS-STING signaling pathway [100]. Accordingly, the inhibition of component of this pathway, such as STING, IRF3, TBK1, resulted in reduced activity of ganciclovir [101].

The correlation between MS and neuroinflammation has been further supported by the involvement of NLRP3 inflammasome in the development of the disease. In fact, it has been reported that activated caspase-1 and IL-1β levels are significantly increased in MS patients and in EAE animal models [102,103,104,105]. Accordingly, mice lacking the expression of inflammasome-involved proteins, such as NLRP3, ASC, and caspase-1, resulted protected from the progression of EAE [106,107]. Moreover, administration of IFN-β weakened the progression of MS by reducing the activity of NLRP3 inflammasome [108]. As reported above, a primary cause of NLRP3 priming is mitochondrial dysfunction. Thus, it is not surprising that mitochondria play a key role in modulating MS progression, as supported by a wide number of studies describing mitochondrial impairments in both MS patients and MS mouse models. Just to cite a few, in MS lesion it has been observed an impaired activity of the electron transport chain (ETC)-enzymes of the complex IV [109], and changes in the aerobic metabolism, mainly due to alteration of mitochondrial superoxide dismutases 1 and cytochrome c levels, were found in platelets of affected patients [110]. Interestingly, these modifications on the ETC are also reflected by alteration of the ROS production, which is increased both in cellular and in animal MS models [111,112,113]. The excessive ROS production exacerbates the oxidative stress resulting in increased mitochondrial lipid peroxidation that leads to the final impairment of mitochondrial activity. The modification in the oxidative process is further boosted by the deregulation of the antioxidant defense mechanism, which has been found altered in MS patients’ body fluids [114,115]. Since the excessive ROS production is also accountable for the induction of DNA mutation, mtDNA sequence variations were found associated with MS [116,117,118]. Lastly, it has been shown that also mitophagy and mitochondrial failure markers are augmented in serum and in CSF samples of MS patients, with a direct correlation with the active phase of the disease [95,119]. All these findings support the hypothesis that the neuroinflammatory process sustains the development of MS, and further highlight the central role of the mitochondria in the progression of the disease.

### 3.2. Parkinson’s Disease

PD is characterized by a progressive loss of dopaminergic neurons in the substantia nigra pars compacta, which is associated to a widespread aggregation of α-Syn forming the Lewy bodies. Accordingly, autosomal dominant mutation of the gene encoding the α-Syn protein (SNCA) determines familial PD [120]. Interestingly, α-Syn, even if predominately localized in neuron terminals, can be found at the mitochondrial surface, where it influences mitochondrial structure and functions [121,122,123]. SNCA is not the only gene responsible for familial PD that directly affects mitochondrial behaviors: to date different biochemical and genetic studies revealed that the production of the PARK genes (indicated in brackets) parkin (PARK2), PINK1 (PARK6), DJ-1 (PARK7), LRRK2 (PARK8), and ATP13A2 (PARK9) are mutated in autosomal recessive Parkinsonism. All these genes work to govern mitochondrial functioning, thus strengthening the evidence that mitochondrial dysfunction is strongly involved in PD development [124].

The idea that mitochondria might be involved in PD arose in the late 1980s, when it was found that the oxidized form of the PD-inducer compound 1-methyl-4-phenyl-1,2,3,6-tetrahydropyridine provoked the inhibition of the complex-I of the ETC in neurons. Accordingly, compromised levels of complex-I were also found in human samples of PD patients [125,126]. Interestingly, neurons from autopsies of PD patients harbored high levels of mutations in mtDNA that correlate with mitochondrial dysfunction [127,128]. The accumulation of mtDNA mutations impairs ETC functioning, leading to compromised Ψm, reduced synthesis of ATP, and increased ROS production. Taken together these results prove the association between pathogenic mtDNA mutations and PD development.

As mentioned above, the NLRP3 inflammasome is the best characterized among the inflammasomes and it has been reported to drive the neuroinflammatory process in PD. In fact, increased expression of inflammasome components and inflammation-related factors have been found in human blood samples of PD patients [129,130]. In addition, the mitochondrial impairment observed in microglia induces an increased ROS production, thus amplifying the NLRP3 inflammasome proinflammatory signaling both in in vitro and in vivo models of PD [131]. Accordingly, it has been shown that the administration of tenuigenin, an anti-inflammatory plant extract, to PD mice models reduces the NLRP3 activation directly acting on ROS production [132]. Moreover, Pink1^−/−^ or Parkin^−/−^ microglia cells have been shown to have an increased NLRP3 activity. This tendency was abolished by the administration of inflammasome inhibitor, both in Pink1^−/−^ or Parkin^−/−^ microglia cells and in patient derived macrophages carrying the PARK2 mutations [133]. The possibility to arrest PD advancement by inhibiting NLRP3 induced neuroinflammation has been confirmed by the administration of MCC950 in rodent PD models, which resulted in a mitigation of motor deficits and reduced accumulation of α-Syn aggregates [134]. Lastly, also carbenoxolone, a heat shock protein inducer, was found to exert beneficial effects in a rat model of PD by inhibiting neuroinflammation and mitochondrial dysfunctions [135].

Ablation of PINK1-parkin pathway, associated with reduced mitophagic process, results in the accumulation of defective mitochondria, damaged mitochondrial proteins and ROS, which leads to NLRP3 inflammasome stimulation. Although this altered pathway was found in PD patient-derived cells and brains the in vivo role of mitophagy in PD remains unclear [136,137,138,139]. Indeed, mice lacking either PINK or parkin do not display PD-relevant phenotypes [140], although in these models a reduced mitophagy pathway was observed [141,142,143]. A recent study has tried to shed light on this aspect by using mitophagy-deficient mouse models with also an increased accumulation of mtDNA mutations, namely Pink1*^−/−^/mutator* and Parkin*^−/−^*/mutator mice. The research performed demonstrates that in these models, acute and chronic stress activate the proinflammatory cGAS-STING pathway leading to the manifestation of dopaminergic neuron loss and movement disorders [144]. Therefore, this work further supports the important connection between mitochondrial stress and inflammation in PD progression demonstrating that mitophagy exerts a crucial role in preventing neuroinflammation in this pathological context.

Taken together all the reported data suggest a tight connection between mitophagy dysfunction, ROS overproduction, and NLRP3 activation, observed in patients affected by Parkinsonism, confirming the fundamental role of mitochondrial driven neuroinflammation in the development of PD.

### 3.3. Alzheimer’s Disease

AD is a NDD with a slow onset that gradually gets worse over time. The main symptom of AD is dementia, which causes problems with memory, thinking, and behaviors, caused by deposition of intracellular neurofibrillary masses of pathologic forms of tau protein and extracellular plaque of Aβ. As described for the other NDDs, mitochondria play a key role also in the pathophysiology of AD. Interestingly, Aβ accumulation was found both in mitochondria of human AD patients’ brains [145] and of transgenic AD mouse models [146]. In detail, it has been shown that Aβ interacts with different mitochondrial components, such as elements of the ETC, diverse mitochondrial matrix proteins, and putative component of the PTP [145,146,147,148]. In this latter scenario, the interaction between Aβ and the PTP component cyclophillin D, induces the pore opening with the consequent alteration of mitochondrial dynamics and functioning (Ca^2+^ homeostasis, ATP levels, ROS) leading to apoptotic neuronal cell death. Notably, PTP is also involved in the regulation of autophagy in AD progression.

The excessive ROS production and the consequent increased oxidative stress is another mitochondrial parameter frequently found dysregulated in AD. Indeed, increased oxidative damage correlates with the brain regions most affected in AD [149,150]. One of the primary targets of oxidative damage is mtDNA, therefore it is not surprising that in AD patient specimens mtDNA mutations are widely present [149,151,152]. As described before, increased ROS production is responsible for inflammasome recruitment, and AD is no exception. Moreover, in AD context, Aβ was found sufficient to activate NLRP3 inflammasome. Accordingly, NLRP3 knockout ameliorated Aβ-related pathology and the development of cognitive decline [153]. Interestingly, mice expressing human tau mutations as well as patients affected by primary tauopathies, such as frontotemporal dementia, exhibited increased NLRP3 levels. Additionally, in this scenario, knockdown of NLRP3 decreased tau aggregation and hyperphosphorylation levels ameliorating the clinical outcome [65]. Similar effects were also obtained by using the specific NLRP3 inhibitor MCC950 in vivo [154]. At demonstration of the determinant role of NLRP3 in AD, elevated levels of its effector molecule IL-1β were found in serum, CSF, and brain of patients with AD as well as other types of dementia [155]. Once secreted, IL-1β enhances the production of Aβ by neurons and induces the phosphorylation of the tau protein [156,157]. Accordingly, IL-1β brain injection upregulates amyloid deposits levels and provokes amyloidogenesis, while IL-1β blockade reduces neuroinflammation by decreasing fibrillar Aβ level and tau activation [158,159]. Polymorphisms of IL-18 promoter, another proinflammatory cytokine released upon NLRP3 inflammasome activation, has been shown to be associated to the risk of developing sporadic late onset AD [160]. Interestingly, IL-18 levels were elevated in body fluids of mild cognitively impaired and AD patients and its production was found elevated in mononuclear cells and macrophages of peripheral blood [161,162]. Furthermore, IL-18 increases the expression of the glycogen synthase kinase 3β and the cyclin dependent kinase 5, which are the mediators of the hyperphosphorylation of tau protein [163].

Proinflammatory cytokines production, including IL-1β, is also enhanced by saturated fatty acid metabolism. Intriguingly, this alternative pathway is the elective way to supply energy in AD brains to overcome the impaired glucose metabolism [164]. Lastly, activation of mitophagy results in diminished Aβ levels and reduced tau hyperphosphorylation leading to a regression of the cognitive impairments in AD-mouse models [165,166]. Accordingly, reduced levels of autophagic and mitophagic markers and an impaired energetic metabolism were observed in human samples obtained by AD-affected patients [167]. The reduced energy supply found in AD patients, due to altered brain glucose metabolism, is compensated by using amino acids and fatty acids as alternative energetic source [168,169]. In conclusion, also for AD development the neuroinflammatory process exerts a pivotal role and represents a powerful therapeutic target.

## 4. Ischemic Stroke and Mitochondrial Induced Neuroinflammation

Ischemic stroke (IS) is a pathophysiological event occurring when the occlusion of cerebral arteries leads to a transient or permanent block of blood supply to a part of the brain [170]. Among the cerebral arteries accounted for in the development of IS episodes, the middle cerebral artery is the most involved. This artery supplies blood to an extended area of the lateral surface of the brain, part of the basal ganglia and the internal capsule, areas that contain motor, sensory functions and emotions [171]. Therefore, depending on the extent of injury, people affected by an IS injury will likely go through a long-term disability or even death [172,173]. Immediately after stroke onset neurons fail to sustain cellular homoeostasis, resulting in a sequence of harmful events strictly connected to mitochondria functions [174]. Mitochondrial failure, triggered by oxygen and glucose deprivation (OGD), leads to neuronal cells damage and ultimately neuron loss. The high energy demand accompanied by limited energy reserves, make neurons the most OGD sensitive brain residing cells [175]. Depending on several factors, including duration of ischemia and circulation in collateral vessels, the failure of blood supply correlates with different outcomes [176,177,178]. The failure of blood supply differentially affects the infarcted brain zone: the infarct core has a low level of reperfusion and is characterized by irreversible damages, while in the penumbra, defined as the damaged but metabolically active neuronal area surrounding the ischemic core, the neuronal structure is still preserved and potentially restorable [176,179]. Notably, even though reperfusion is a mandatory step to recover ischemic damage, ischemic reperfusion (IR) is a double-edged sword. If on the one hand IR is a key factor in safeguarding the lesioned brain tissue, on the other hand it establishes the IR-injury exacerbating brain damage [177,180,181].

Recently, a central role of cytosolic dsDNA-sensing cGAS in sterile inflammation and following ischemic injury has been reported [182]. Briefly, in the middle cerebral artery occlusion in vivo model of IS, it has been found that pharmacological inhibition of dsDNA cGAS by A151, a selective antagonist, reduced microglia activation within the ischemic penumbra, inhibited the release of proinflammatory cytokines and reduced the migration of periphery neutrophils injury improving ischemic outcome. Furthermore, AIM2 inflammasome implicated in the brain damage and neuroinflammation after IS [183,184] is also inhibited by A151 [182]. In line with this evidence, CX3CR1CreER mice carrying the selective deletion of cGAS in microglia were protected from ischemic injury [182], suggesting that inhibition of dsDNA sensing cGAS could represent a promising target against IS injury.

A large number of works aimed to define the crucial targetable pathways involved in IS, especially to overcome the detrimental effects of reperfusion, identified NLRP3 mediated neuroinflammation as an eligible target [185,186,187,188]. Although the expression of different inflammasome components including NLRP1, NLRP3, NLRC4, ASC, caspase-1, and the proinflammatory cytokines IL-1β and IL-18 increases in the initial hours and early days after ischemia in the brain of rodents [183,189], mitochondrial destabilization or dysfunctions are tightly associated with only NLRP3 inflammasome activation [61,190,191].

The main causes of NLRP3 inflammasome activation in ischemic conditions are ascribable to mitochondria dysfunction, as abnormal Ca^2+^ influx, ROS production, mitochondrial membrane permeabilization with the consequent release of DAMPs and mtDNA, all conditions that are tightly linked one to the other (Figure 3). Within minutes, OGD affects mitochondria functions leading to a strong reduction of OXPHOS and therefore of ATP synthesis. Neurons in the infarct core fail to compensate the ATP cell request leading to a bust of the Na^+^/K^+^ ATPase pump [185,192]. This results in neuronal membrane depolarization accompanied by an extreme release of glutamate in the extracellular space finally leading to neurons loss [193]. Glutamate is an important excitatory neurotransmitter, which binds several types of receptors such as N-methyl-D-aspartate (NMDA) receptor, a-amino-3-hydroxy-5-methyl-4-isoxazolepropionic acid receptor and kainate receptor [194]. Although these receptors originally were thought to be exclusive to neurons, several studies revealed their functional expression also on glial cells [195,196]. The excessive amount of glutamate after stroke leads to a hyperactivation of these receptors promoting a strong influx of Ca^2+^ into the cells [197,198,199]. Mitochondria are the crucial players in regulating cytosolic Ca^2+^ levels [200], thus the massive accumulation of cytosolic Ca^2+^ results into the activation of the mitochondria calcium uniporter (MCU) leading to mitochondrial depolarization, which in turn drives NLRP3 inflammasome activation and the consequently IL-1β release [201]. Consistently, it has been found that in response to the activation of NMDA receptors, MCU overexpression increases mitochondrial Ca^2+^ levels and provokes mitochondrial membrane depolarization. Inversely, genetic knockdown of MCU reduces the NMDA-induced increase in mitochondrial Ca^2+^ followed by lower levels of mitochondrial depolarization [202]. In line with these findings, in focal cerebral ischemia rat models, the early stages of cerebral ischemia are characterized by an upregulation of the mitochondrial calcium uptake 1 (MICU1) a crucial regulator of MCU [203]. Interestingly this occurs in the acute phase of IS right when the inflammatory response takes place [204,205,206]. To corroborate the role of Ca^2+^ in mitochondria dysfunction-induced inflammation, it has been shown that NLRP3 inflammasome activation is reduced following the inhibition of extracellular Ca^2+^ entry or the depletion of Ca^2+^ stores in the ER [207]. Albeit K^+^ efflux, a common NLRP3 inducer [208], has been proposed to be upstream of the Ca^2+^-induced NLRP3 inflammasome activation, thus indicating that high levels of extracellular K^+^ abolish NLRP3 activation [209], the crucial contribution of mitochondrial Ca^2+^ overload in sustaining inflammasome activation following IS and IR is not excluded.

Additionally, in brain ischemic damage, mitochondria and ROS have a crucial role [210,211]. Following cerebral ischemia, the balance between ROS production and clearance is compromised, resulting in a pathogenic oxidative-stress-induced inflammation signaling. After IR, the spreading of mitochondrial activity results in a burst of ROS levels [212], worsening the inflammatory response and then the ischemic outcome. Mitochondrial ROS are predominantly generated by complexes I (NADH dehydrogenase) and III (cytocrome bc) of the ETC. Indeed, free electrons in the mitochondrial ETC leaking out and reacting with molecular oxygen, generate O_2_^•−^ as a metabolic by-product of respiration [213,214,215]. Recently, complex I has been distinguished as a major source of ROS upon IR. Briefly, following IR the succinate, which is markedly increased during ischemia, becomes oxidized. By reverse electron transport, the oxidized metabolite promotes ROS formation at the complex I, providing the initiating burst of O_2_^•−^ that leads to IR injury [216]. In agreement, the treatment with rotenone, a mitochondrial complex I inhibitor, causes the loss of Ψm and thus increases ROS production, enhancing NLRP3-dependent IL-1β secretion [61,217]. Ca^2+^ accumulation and ROS production, during IR, lead to the mitochondrial PTP induction. The opening of PTP allows the release of mitochondrial material to the cytoplasm including DAMPs, such as cardiolipin and mtDNA [218]. The ability of cyclosporin and other PTP inhibitors to attenuate NLRP3 inflammasome activation provides a link between PTP and inflammation [219,220]. Several works reported that the inhibition of PTP by genetic or pharmacological approaches confers protection against ischemic damage [221,222,223]. Accordingly, it has been observed that ROS generation induced by rotenone injection, are mitigated by the inhibition of PTP or mitochondria ROS scavenger [224], indicating the important participation of PTP in the activation and sustainment of IS-induced inflammation.

Mitophagy exerts a fundamental role in cerebral ischemia by preventing all the described mitochondrial-dependent proinflammatory processes, such as extreme ROS production, loss of mitochondrial membrane polarization, and PTP opening [225,226,227]. Indeed, it has been reported that melatonin administration promotes inhibition of both ROS generation and NLRP3 inflammasome activation by increasing mitophagy [228]. Moreover, both methylene blue administration and rapamycin treatment have been found to enhance mitophagy, reducing ROS accumulation and mitochondrial dysfunction following cerebral ischemia [229,230]. Lastly, the overexpression of the activating transcription factor 4 has been found to ameliorate cerebral IR by suppressing NLRP3 inflammasome activation through parkin-dependent mitophagy [231]. By contrast, knockout of BNIP3-like (BNIP3L), an important player in cerebral IR-induced mitophagy, worsen cerebral IR injury in mice causing an impairment in mitophagy, condition that could be rescued by BNIP3L overexpression [232]. However, it has also been reported how excessive mitophagy could be detrimental for both the ischemic and the reperfusion state, even though the precise molecular pathway has still to be defined [233,234]. Given the importance of mitophagy in maintaining mitochondrial homeostasis, and therefore inhibiting also NLRP3- and ROS-induced neuroinflammation, the modulation of this catabolic process in IS represents an important therapeutic target.

## 5. The Neuroinflammatory Process in Epilepsy: The Involvement of Mitochondria

Epilepsy is a progressive neurological disorder affecting almost the 1% of the global population, characterized by recurrent seizures and by other complex features, including psychiatric and cognitive comorbidities [235,236]. More than 30% of epileptic patients are drug resistant, becoming thus affected by refractory epilepsy, which can be progressive. Epilepsy and its comorbidities can lead to a profound deterioration in the patient’s quality of life [237]. Recent studies conducted on animal models have highlighted that neuroinflammation plays a crucial role in precipitating and/or sustaining seizures recurrence, ultimately facilitating neural cell loss [238]. Indeed, intracerebral application of interleukin IL-1β has been shown to increase seizure activity in experimental models [239]. The oxidative stress generated by RNS and ROS imbalance, due to mitochondria dysfunction, can lead to alterations of cellular macromolecules, such as lipids, proteins, and DNA, [240] with the consequent generation of “oxidation specific epitopes” that induces neuroinflammation [241,242] (Figure 4). Mitochondrial dysfunction is one of the prominent pathological hallmarks that aggravates the inflammation process during epilepsy. Approximately 40% of the epileptic patients have been reported to be affected by a mitochondrial disease [243].

Temporal lobe epilepsy (TLE) is an acquired epilepsy, usually triggered by an insult (such as brain injury) that leads to the development of spontaneous, recurrent seizures after a latency period of months to years [244]. This latency period corresponds to an “epileptogenesis” process, in which a normal brain is transformed to one capable of generating spontaneous seizures [245].

It has been reported that inflammasomes activation contributes to the development and progression of epilepsy through the release of inflammatory mediators [246]. Remarkably, in vivo injection of NLRP3 small interfering RNAs displayed neuroprotective effects in rats following amygdala kindling-induced epilepsy [247]. Moreover, it has been speculated that amentoflavone, a natural biflavone compound with anti-inflammatory and antioxidative properties, has the ability to affect epileptogenesis and exerts neuroprotective effects through the inhibition of the NLRP3 inflammasome [248]. Interestingly, IL-10 administration in the picrotoxin seizure model results in a decreased activation of the NLRP3 inflammasome, thus of IL-1β release, suggesting a protective role in status epilepticus (SE) [249]. The role of NLRP3 inflammasome in the development of epilepsy was confirmed by a recent study showing that children with febrile seizures have higher serum levels of IL-1β, correlated to NLRP3 upregulation in PBMCs, as compared to healthy controls [250]. Recent studies have demonstrated that, like NLRP3, also the NLRP1 inflammasome is involved in SE. In fact, a NLPR1 polymorphism was reported in a Chinese Han population affected by partial seizures, suggesting a broader role of inflammasomes in inducing vulnerability to seizures [251]. Moreover, it has been demonstrated that the expression of NLRP1 and caspase 1 were increased in the hippocampus of individuals with pharmacoresistant mesial TLE, compared with the control group [252]. Accordingly, knocking down NLRP1 expression in TLE rats led to decreased hippocampal neuronal loss and reduced seizure frequency and severity [252]. Finally, analysis of the hippocampal tissue transcriptome of patients affected by mesial TLE demonstrated an upregulation of NLRP1 compared to non-epileptic controls [253].

Besides the important role of inflammasomes activation in epileptogenesis, several studies in both humans and rodent models of TLE and SE suggest a close association between mitochondrial dysfunction and oxidative stress [254,255,256]. Mitochondrial respiratory deficit occurs during experimental TLE, and ROS production contributes to this event [257]. In the kainic acid model, a commonly used model of epilepsy associated with neuronal death, an increase in mitochondrial O_2_^•−^ and of 8-hydroxy-2-deoxyguanosine levels, an indicator of oxidative DNA damage, have been observed. The intracerebroventricular infusion of the catalytic antioxidant MnTBAP 48 h before kainic acid injection has been reported to significantly reduce neuronal cell death [258]. These results confirm the strict association between mitochondria-mediated inflammation and seizure induced neuronal loss. The role of ROS in modulating seizure-induced neuroinflammation was also investigated in the pilocarpine model of TLE [240] showing that the injection of MnIIITDE-2-ImP5+, a catalytic scavenger of O_2_^•−^, attenuated SE-induced microglial activation, mitochondrial dysfunction, and hippocampal neuronal loss. Moreover, MnIIITDE-2-ImP5+ improved short- and long-term recognition memory as well as spatial memory in epileptic rats even after treatment discontinuation. Moreover, there was no positive effect in terms of spontaneous seizures, suggesting that learning and memory improvements were not due to a reduction of the overall seizure burden [259]. The occurrence of repeated seizures leads to the activation of NMDA receptors that, as described above, induces strong influx of Ca^2+^ into the cytoplasm. Increased cytoplasmic Ca^2+^ concentration results in the activation of MCU, thus increasing mitochondrial Ca^2+^ levels that trigger the activation of various enzymes, such as nitric-oxide synthase, calpains, and NADPH oxidase, leading to the progressive inflammation [260]. Indeed, a prolonged seizure-like activity increases ROS production in an NMDA receptor-dependent manner in glioneuronal cultures, and this activity can be reduced with the inhibition of NADPH oxidase or xanthine oxidase [261]. Accordingly, the administration of a NMDA receptor antagonist after in vivo SE provided significant neuronal protection [262]. In a clinical study of parahippocampal and hippocampal tissue samples from 74 mesial TLE patients, mitochondrial dysfunction due to ROS-mediated mtDNA mutagenesis has been shown to promote neuronal cell death and epileptogenesis [263]. Accordingly, inducible NO synthase inhibition, and therefore the reduction of peroxynitrite production, may alleviate neuroinflammation and represent a neuroprotective strategy against SE [260]. Increased ROS lead to the accumulation of damaged mitochondria in the brain, which are normally removed by mitophagy [264]. Interestingly, it has been found that mitophagy is highly active in samples from hippocampi and temporal lobe cortices obtained from patients with refractory TLE, but it is unable to remove damaged mitochondria completely, thereby favoring neuronal death [265]. In this scenario, incomplete mitophagy correlates with TLE pathology. On the other hand, the treatment with DA3-CH, a glucagon-like peptide-1 and glucose-dependent insulinotropic polypeptide receptor agonist, has a neuroprotective effect in the pilocarpine model of epilepsy, because of its ability to attenuate mitophagy and, consequently, neuronal death [266]. Furthermore, the inhibition of succinate dehydrogenase in kainic acid or pilocarpine-induced SE results in a decrease of succinate levels, oxidative stress, and mitophagy, preventing neuronal damage and reducing severity of epileptic seizures [267]. Therefore, mitophagy appears to play a double faceted role in epilepsy, and the conditions in which it has a protective or pathogenic role are still controversial. In conclusion, some controversies notwithstanding, the contribution of mitochondria in epilepsy is likely important, and may lead to identification of conceptually new therapeutic approaches.

## 6. Current Therapies Targeting Neuroinflammation

As largely described above, mitochondria have a great impact on the neuroinflammatory process that is beyond the development of different brain disorders, such as NDDs, epilepsy, and IS, but, from a therapeutic point of view, directly targeting mitochondria is still a complicated route. In fact, current therapies aim to treat neuroinflammation with a wider approach. One of the most characterized proinflammatory molecules is the IL-1β, and its impact on seizures occurrence was described more than 20 years ago when Redman and collaborators registered neurotoxicity after daily administration of 50 ng/kg i.v. IL-1β in patients with metastatic renal cells carcinoma [268]. Since then, many steps forward were made to understand the inflammation process and its neurotoxic contribution.

### 6.1. Targeting Neuroinflammation in Multiple Sclerosis

MS is characterized by a demyelinating autoimmune nature suggesting the promising results of therapies aimed to reduce inflammation. Patients affected by MS lack the ability to complete a successful remyelination process after the progression of demyelination [269]. In particular, inefficient clearance of myelin debris seems to play a crucial role in preventing a proper remyelination [270]. In this context, the lack of CX3C chemokine receptor 1 (CX3CR1), was described to compromise remyelination in mouse models [270]. In CX3CR1-deficient mice, the clearance of myelin debris was blocked, obstructing the correct remyelination. These data highlighted the crucial role of microglia in the clearance of myelin debris after a primary demyelinating insult. Additionally, IFN-β secreted by microglia enhanced the removal of myelin debris in the MS model of experimental autoimmune encephalomyelitis (EAE) [271]. These data indicate that remyelination cannot be successful if myelin debris are still present, thus pointing to the myelin clearance process as a promising target. Besides the great importance of proinflammatory cytokines, such as IFN-β, in modulating myelin debris clearance, excessive inflammatory response is detrimental in MS development, therefore different therapeutic approaches aim to decrease the neuroinflammation in MS. In particular, an interesting strategy to counteract immune system response is provided by alemtuzumab. Patients treated with this compound showed a reduction of proinflammatory cytokines. Furthermore, alemtuzumab showed a long-lasting therapeutic benefit with the production of anti-inflammatory cytokines such as IL-10 and transforming growth factor β (TGF-β) [272]. However, the presence of secondary autoimmunities caused by alemtuzumab has limited its clinical employment [273]. Among the drugs with neuroinflammatory effects, FDA has recently approved ocrelizumab which targets CD20-positive B-cells and prevents damage to nerve cells. There are many ongoing phase III/IV clinical trials evaluating the dose efficacy of ocrelizumab by recording the improvement of ambulatory functions (NCT04544436; NCT04387734). Hoffmann-La Roche has also started a phase III trial of fenebrutinib, a dual inhibitor of both B-cells and myeloid lineage-cells activation, in patients with primary progressive MS (NCT04544449). Similarly, ofatumumab, a human anti-CD20 antibody responsible for specific B-cells lysis and depletion at the lymph nodes, is undergoing phase III clinical trial (NCT04486716). Moreover, the Food and Drug Administration has approved the use of fingolimod to treat relapsing-remitting MS. Fingolimod is a sphingosine-1-phosphate-receptor modulator that blocks the recirculation of autoaggressive lymphocytes without suppressing the immune response [274]. Various clinical trials have confirmed positive effects conferred by fingolimod treatment with lower relapse rate and better magnetic resonance imaging outcome [275,276].

### 6.2. Targeting Neuroinflammation in Parkinson’s Disease

High levels of IL-1β were reported in Parkinsonian patients [277], thus many studies have investigated the contribution of inflammation to the onset of PD. In particular, the focus has pointed to the IL-1 receptor/Toll-like receptor 4 axis, as the trigger of the neuroinflammatory process [278]. However, therapies specifically targeting this pathway are still missing and are mainly focused on PD symptoms. Among these therapies, L-dopa still represents the most effective symptomatic pharmacological treatment for PD [279]. Despite this, several strategies have been recently studied to modulate the inflammatory response. For instance, minocycline, a widely used antibiotic, seems to protect nigral dopaminergic neurons and to decrease glial cells activation [280], warranting its neuroprotective role in PD. Unfortunately, this treatment is still controversial because, despite its compelling features, it has also been reported to cause severe neural cell loss in animal models [281]. Besides minocycline, other compounds have been described to have a neuroprotective effect in the treatment of Parkinsonism. For instance, the synthetic steroid dexamathasone prevents dopaminergic neurons degeneration in a mouse PD model [282], and the synthetic compound naloxone blocks microglia activation thus reducing the inflammatory damage [283]. Aside from its cardiovascular application, nimodipine, a calcium channel blocker, exhibits a neuroprotective effect on dopaminergic neurons by downregulating microglial activation, thus decreasing TNF-α and IL-1β production [284]. Another compound with promising effect in targeting neuroinflammation in PD is semaglutide, a synthetic analogue of glucagon-like peptide 1 (GLP1), which stimulates GLP1R, and it is already used to treat type 2 diabetes. GLP1R activation inhibits the production of proinflammatory cytokines and slows down the neurodegenerative process [285]. An ongoing phase II clinical trial will assess the effects of semaglutide on both motor and non-motor symptoms of PD (NCT03659682). Another GLP1 agonist, exenatide, is undergoing a phase III clinical trial to verify its neuroprotective effect in Parkinsonian patients (NCT04232969). As previously described, oxidative stress has a crucial role in triggering the degeneration of dopaminergic neurons. Accordingly, studies on different antioxidant therapies, such as idebenone a quinone analogue, and tocotrienols, primary form of vitamin E, are undergoing, respectively, phase II and phase IV clinical trial for their effect on motor and non-motor symptoms in patients with early PD (NCT03727295, NCT04491383).

### 6.3. Targeting Neuroinflammation in Alzheimer’s Disease

Nimodipine together with the non-steroidal anti-inflammatory drugs (NSAIDs) are the elective drugs used to treat neuroinflammation in AD. Nimodipine prevents Aβ-dependent injury, and its administration both in in vitro and in vivo models resulted in a strong inhibition of IL-1β release and accumulation [286]. Ibuprofen, which is one of the most prominent NSAIDs, is involved in many ongoing clinical studies for AD treatment, but its action is still controversial. In fact, a clinical study reported that long NSAID treatment reduced the probability of developing AD [287], while on the other hand another one reported that AD patients treated with NSAIDs have a worse outcome [288]. Recently, fenamate NSAIDs were reported to have beneficial effects in a mouse model of AD [289], while a pioglitazone phase III clinical trial was terminated because of lack of efficacy [290]. Chronic treatment with AF710B (ANAVEX 3-71), a selective allosteric M1 muscarinic and sigma-1 receptor agonist, showed anti-amyloid and anti-neuroinflammatory effects suggesting its ability in damping inflammation in animal models of AD [291]. Following the encouraging preclinical results, a phase I clinical trial is assessing tolerability and safety of ANAVEX3-71 compound in healthy volunteers (NCT04442945). The same pharmaceutical company developed another similar compound, ANAVEX2-73, which is now ongoing phase III clinical trial to evaluate its anti-amyloid and anti-inflammatory effects on AD patients (NCT03790709). Primary outcome measures will be available in the next few years. Nilvadipine, a calcium channel blocker, due to its anti-inflammatory potential shown in preclinical studies, underwent a phase III clinical trial to treat AD. Unfortunately, the trial reported no benefits of nilvadipine in the treatment of mild to moderate AD (NCT02017340). Lastly, epidemiological studies proposed the treatment of AD with NSAIDs, but the encouraging results recorded in animal models have not yet been confirmed in human patients [292].

### 6.4. Targeting Neuroinflammation in Ischemic Stroke

As observed in MS, the oral administration of fingolimod displayed promising results in ameliorating acute IS outcome [293]. Treated patients exhibited a reduction of lesions and a better clinical recovery after 3 months. As well as for PD, minocycline showed controversial results in stroke patients: while a phase II trial reported a good outcome, a pilot study has shown that minocycline treatment was safe but not efficacious [294,295], indicating the need of further investigation. The recombinant human IL-1 receptor antagonist, namely anakinra, showed encouraging results in a phase II clinical trial by decreasing the neuroinflammatory process after 3 months from the administration, even though this promising result was not reflected by the clinical outcome [296,297]. A novel combined therapy of molecular hydrogen H_2_, an antioxidant, plus minocycline, named H_2_M is undergoing a pilot randomized control trial to test its efficacy in preventing brain tissue damage (NCT03320018). Lastly, the efficacy of the neuroprotectant nerinetide is being investigated in patients with acute IS after the promising results obtained in vitro showing a reduction of the hypoxic damage in cultured neurons mimicking IR injuries [298] (NCT04462536).

### 6.5. Targeting Neuroinflammation in Epilepsy

As well as for PD, toll-like receptor 4 has been described to significantly affect the neuroinflammatory process and the pathogenesis of epilepsy [299,300]. Therefore, IL-1R has become a front-runner among the possible therapeutic targets. Accordingly, promising results, such as the reduction of total number of seizures, have emerged from the use of anakinra to treat epilepsy and related syndromes [301]. Interestingly, levetiracetam, one of the most commonly used anti-epilepsy drugs, reported anti-inflammatory effects in vitro [302]. Cannabidiol has also been investigated for both its anti-inflammatory and antioxidant properties [303]. Even though different phase III clinical trials are currently assessing the efficacy of cannabidiol in decreasing the number of seizures, FDA has recently approved epidiolex, a pharmaceutical compound containing cannabidiol, for the treatment of seizures associated with two severe forms of epilepsy (NCT02224690) (see Table 1).

## 7. Conclusions

Neuroinflammation has been shown to play a pivotal role in CNS disorders, being mainly responsible for the neuronal cell loss and the exacerbation of the pathology. As widely described, mitochondria have a prominent part in this process: they are responsible for inflammasome assembly and ROS production, and they enclose a large amount of DAMPs, accountable for sustaining the inflammatory process. Interestingly, all these proinflammatory roles are balanced by mitophagy, which is responsible for eliminating abnormal mitochondria in order to interrupt the inflammatory sprouts. These aspects are finely balanced in CNS physiological homeostasis, since acute inflammation cover also a protective role in case of pathogens infection, but the minimal perturbation of this fine regulation triggers the neurodegenerative process. As neurodegeneration involves many altered pathways, researchers have focused the attention on many possible targets including abnormal oxidative stress and uncontrolled neuroinflammation. Researchers are investigating why pathways that are crucial for cell survival are dysregulated in neurodegenerative diseases. For instance, neuroinflammation is vital in activating the regenerative process in IS and epilepsy, but its atypical stimulation leads to opposite results. The actual used therapies act mainly on symptomatic relief, and different pharmacological mechanisms of used drugs are yet to be fully elucidated. At the same time, even if many preclinical studies reported encouraging results, various clinical trials have shown controversial outcome or poor efficacy. This scenario could be ascribed to one of the biggest challenges of clinical trials, that is patients’ heterogeneity and the presence of comorbidities. For these reasons, as far as now, no therapies have shown efficacy in preventing neurodegenerative diseases or at least in significantly reducing their progression. However, an increasing number of studies are focusing on deregulated neuroinflammatory pathways as a common feature in NDDS. In particular, in MS, due to its autoimmune nature, drugs targeting neuroinflammation have already been approved and have displayed the most prominent results.

In conclusion, even if many aspects of inflammation-driven neurodegeneration are still unclear and further studies are needed to exploit all the pathways beyond this phenomenon, dysregulated inflammatory responses appear as a common feature for progression of brain diseases. Therefore, targeting neuroinflammation, also by acting on mitochondria, in NDDs, IS, and epilepsy represents a promising complementary therapy to obtain better clinical outcomes.

## Figures and Tables

**Figure 1 biomolecules-10-01437-f001:**
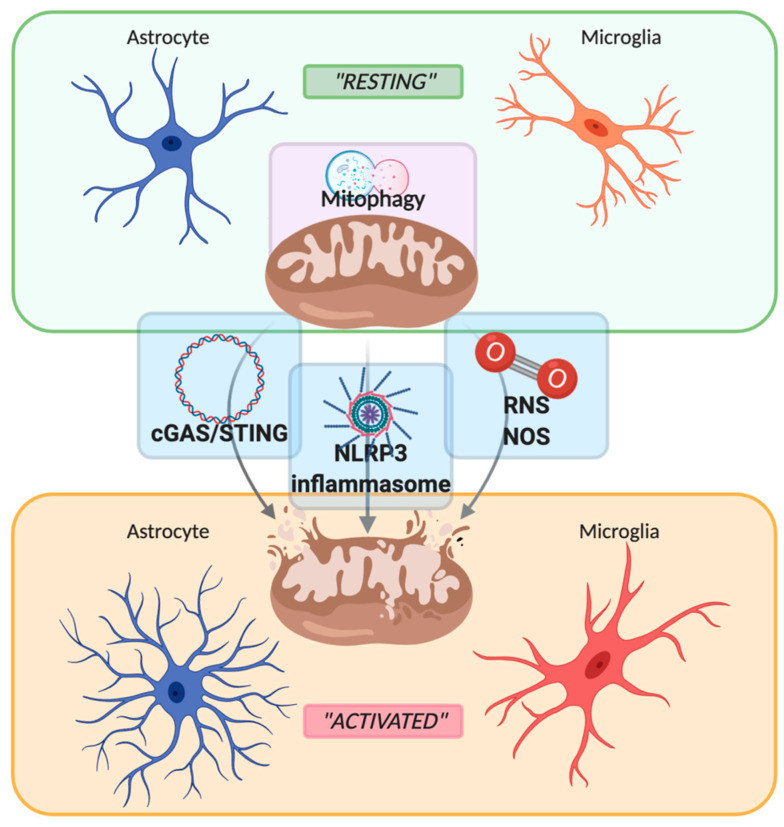
Schematic representation of the switch between “resting” and “activated” state of astrocyte and microglia mediated by mitochondria. Mitochondria maintain their healthy and physiological state by mitophagy. Upon stressful condition, such as inflammatory stimuli, mitochondria are disrupted with the consequent release of mtDNA, damage-associated molecular patterns (DAMPs), reactive oxygen/nitrogen species (ROS and RNS), leading to the activation of the sentinel of the central nervous system such as astrocytes and microglia. Created with BioRender.com.

**Figure 2 biomolecules-10-01437-f002:**
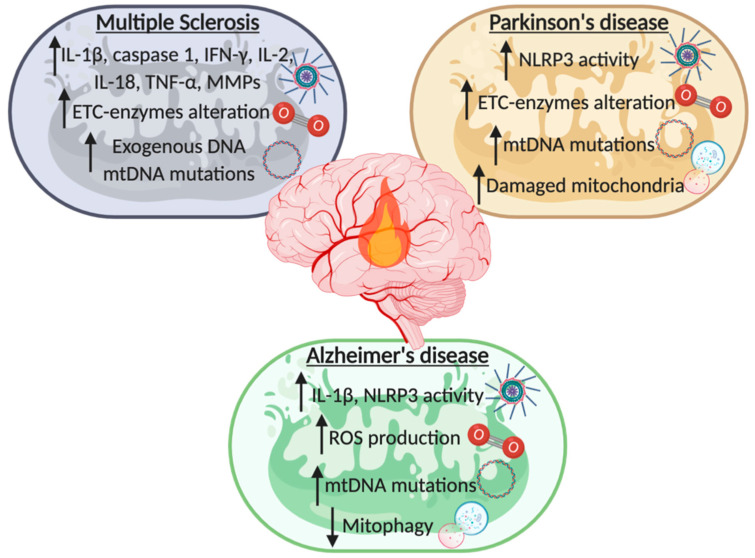
Schematic list of mitochondrial contribution to the recurrence of neuroinflammation during the development of neurodegenerative diseases (NDDs). In multiple sclerosis, Parkinson’s disease, and Alzheimer’s disease mitochondria have a fundamental role in the induction of a neuroinflammatory state by the activation of inflammasomes, variation of the electron transport chain (ETC) enzymes, modulation of reactive oxygen species (ROS) production, alteration of the mtDNA and of the mitophagic process. Created with BioRender.com.

**Figure 3 biomolecules-10-01437-f003:**
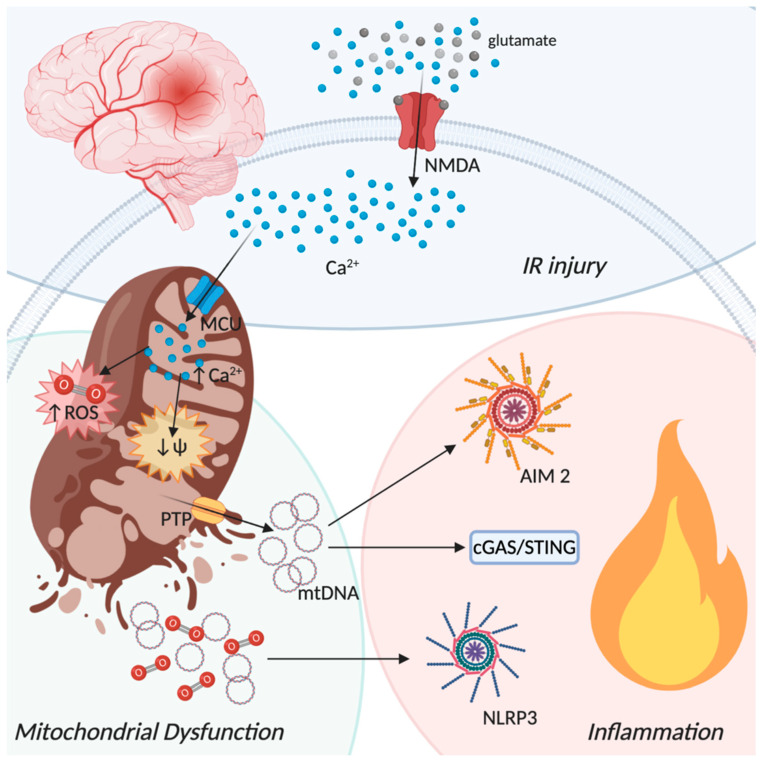
Ischemic-reperfusion injury from a mitochondrial perspective. Oxygen depletion occurring during an ischemic event lead to neuronal membrane depolarization with glutamate release in the extracellular space. Glutamate binds to N-methyl-D-aspartate (NMDA) receptors promoting a strong influx of Ca^2+^. The increase of cytosolic Ca^2+^ activates the mitochondria calcium uniporter (MCU) leading to mitochondrial depolarization and increased production of reactive oxygen species (ROS) with the consequent opening of the permeability transition pore (PTP) and the final disruption of the mitochondrial membranes. mtDNA and ROS release in the cytoplasm drives the inflammatory process by the activation different pathways, such as AIM2 and the NLRP3 inflammasomes and the cGAS-STING dsDNA sensing machinery. Created with BioRender.com.

**Figure 4 biomolecules-10-01437-f004:**
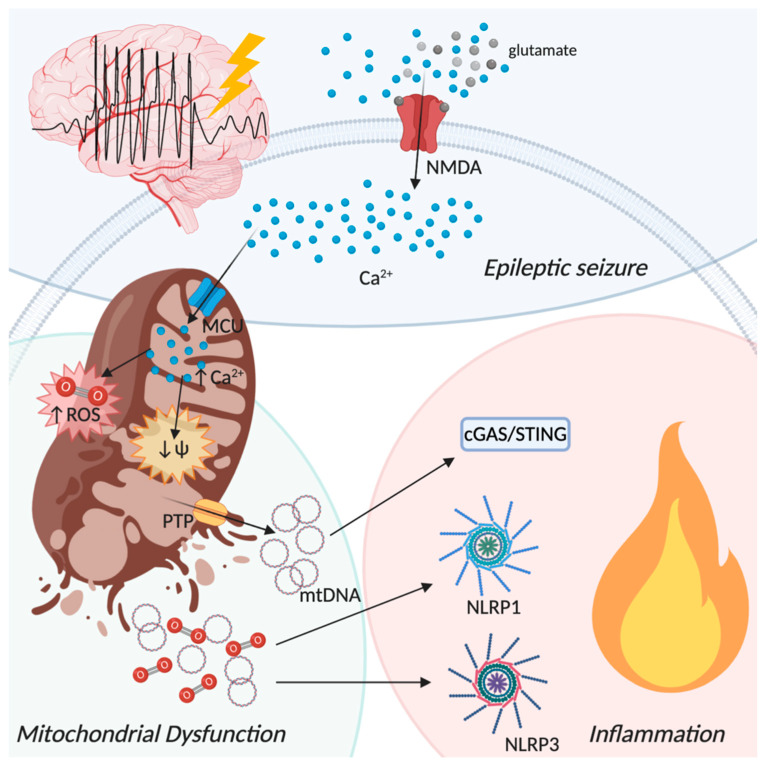
Epileptic seizure from a mitochondrial perspective. Repeated epileptic seizures activate NMDA receptors with the consequent influx of Ca^2+^ into the cytoplasm. Additionally, in this scenario, increased cytoplasmic Ca^2+^ concentration activates MCU. Augmented mitochondrial Ca^2+^ level triggers the activation of different enzymes responsible for reactive oxygen species production (ROS), such as nitric-oxide synthase and NADPH oxidase, and leads to membrane depolarization with the opening of the permeability transition pore (PTP) and mitochondrial membrane disruption. Release of mitochondrial components in the cytoplasm activates proinflammatory pathways such as NLRP1 and NLRP3 inflammasome assembly and the cGAS-STING dsDNA sensing machinery, leading to progressive inflammation. Created with BioRender.com.

**Table 1 biomolecules-10-01437-t001:** List of recent studies and drug therapies targeting neuroinflammation in Parkinson’s disease, Alzheimer’s disease, multiple sclerosis, ischemic stroke, and epilepsy. The clinically approved column refers to the approval of the drug for the treatment of the specific CNS disease. Refer to the text for the explanation of controversial effects.

Therapy	Disease	Effects	Clinically Approved	Reference
Alemtuzumab	MS	Controversial	No	[272,273]
ANAVEX2-73	AD	Anti-inflammatory; Antioxidant	No	NCT03790709
ANAVEX3-71	AD	Anti-inflammatory	No	[291]; NCT04442945
Cannabidiol	Epilepsy	Anti-inflammatory; Antioxidant	Yes	[303]; NCT02224690
Dexamethasone	PD	Neuroprotectant	No	[282]
Exenatide	PD	Neuroprotectant	No	NCT04232969
Fenamate NSAIDs	AD	Anti-inflammatory	No	[289]
Fenebrutinib	MS	Anti-inflammatory	No	NCT04544449
Fingolimod	IS	Neuroprotectant	No	[293]
Fingolimod	MS	Anti-inflammatory	No	[274,275,276]
H2M	IS	Antioxidant; neuroprotectant.	No	[287,288]
Ibuprofen	AD	Controversial	No	[287,288]
Idebenone	PD	Antioxidant	No	[299,300]
IL-1Ra	Epilepsy	Anticonvulsant	No	[299,300]
IL-1Ra	IS	Controversial	No	[296,297]
Levetiracetam	Epilepsy	Anticonvulsant; anti-inflammatory	Yes	[302]
Minocycline	IS	Controversial	No	[294,295]
Minocycline	PD	Controversial	No	[280,281]
Naloxone	PD	Neuroprotectant	No	[283]
Nerinetide	IS	Neuroprotectant	No	[298]; NCT04462536
Nilvadipine	AD	Controversial	No	NCT02017340
Nimodipine	AD	Anti-inflammatory	No	[286]
Nimodipine	PD	Neuroprotectant	No	[284]
Ocrelizumab	MS	Neuroprotectant	Yes	NCT04544436; NCT04387734
Ofatumumab	MS	Anti-inflammatory	No	NCT04486716
Pioglitazone	AD	Controversial	No	[290]
Semaglutide	PD	Anti-inflammatory	No	[285];NCT03659682
TLR4 deletion	PD	Neuroprotectant; Anti-inflammatory	No	[278]
Tocotrienols	PD	Antioxidant	No	NCT04491383

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
