# Peer review of "The Dichotomous Role of Inflammation in the CNS: A Mitochondrial Point of View"

_biomolecules, 2020, doi:10.3390/biom10101437_

Round 1

Reviewer 1 Report

General Remarks

  1. Language is proficient. Minor grammar corrections are needed especially in the second half of the text.
  2. The broader view on the mitochondrial function is missing. The information on mitochondrial function and different aspects of its role in inflammation is fragmented while few chosen protein molecular pathways are described in detailed way.
  3. The significant bias is present with focusing on extreme pathological situation, which could be understood by less experienced reader as a normal situation. It should cover both in health and in sickness comparison to present an honest point of view and appropriate perspective and proportions of the mechanisms.
  4. The microglia and astrocytes are mainly taken into the account – often intermixed and difficult to understand which cells are discussed in given aspect. The role of neurons is narrowed to their death.
  5. The idea of “mitochondrial induced neuroinflammation” is over simplified and biased.
  6. Perspectives – authors show only few drugs for future perspectives but do not point out the main directions for future studies. The realistic point of view of pros and cons - if antioxidant or anti-inflammatory drugs actually do have potential for disease treatment is missing.
  7. The text is chaotic. Authors suddenly jump between subjects and it is difficult to follow the main thought in some parts, potentially misleading. A systematization of the information within and between paragraphs would help.
  8. Length – shortening the text by removing repetitions, especially in the first half of the text would help in the perception of the data.

Minor remarks:

  1. Abstract -
  2. Introduction:
  3. Line 43 – about the oligodendrocytes – I suggest to add something like “as far as we are now aware the oligodendrocytes do  not  have  a  striking  role  in  the  physiological  neuroinflammation…”.
  4. Line 49 – about astrocytes – I suggest to also mention the local energy metabolism regulation as the very important and mitochondria related function.
  5. Line 64 – “performed” or other word instead of “exploited”
  6. Before focusing on the chronic neuroinflammation and its damaging role I suggest to write a sentence setting up the baseline for the process. Something like: “The neuroinflammation is a natural process of defence, precisely and timely regulated, with both a pro-inflammatory phase neutralizing the danger and anti-inflammatory phase restoring the homeostasis and regenerative.” From there you can easily indicate that the prolonged or very strong neuroinflammation is dangerous because regulation of the natural process is distorted. Pro-inflammatory factors in adequate amounts are also required for healthy functioning, just like ROS. The issue is in the balance between pro- and anti-inflammation. I think this is worth underlying in the text.
    -There is a similar sentence in line 75 – I suggest moving it earlier.
  7. Line 86 – although I understand what you meant by this sentence it is potentially misleading. Can you add an example of such functions there (sensing cellular damage, promoting response).
  8. Line 106-109 – placing astrocytes at this point of the story gives false impression that astrocytes are the main PRR expressing cells, while those are microglia. Astrocytes also express PRR but to a minor extent and this proportions should be clearly stated and supported by appropriate reference. Astrocytes mostly rely on microglia on danger recognition and it’s difficult to activate them when there is no microglia around.
  9. Line 121 – the definition of inflammasome should be provided.
  10. Line 142 – OMM was not developed in any particular cause. It just has particular properties. Please be careful with such formed sentences suggesting some form of aim-oriented intervention.
  11. Line 143 – there is a mix-up of substances released from mitochondria vs receptors triggered on mitochondria. Please clear this issue. When first mentioning PRR (line 109) it should be stated where is it expressed intra- or extracellularly.
  12. It would be helpful to list the examples of PRRa (divided to extra- and intra-cellular) and DAMPs, PAMPs – in a table, text or graph.
  13. Can authors somehow differentiate the predominating functions of mitochondria between microglia, astrocytes and neurons? How their mitochondria differ?
  14. Line 226: “When the  redox  state  is  balanced,  ROS  act  as  repairers  of  cell  damage,  contributing  to  the  conservation  of  cellular  functionality” – this sentence is wrong. How reactive oxygen species can repair cells?
  15. Line 227: “However, when  oxygen  homeostasis is not maintained, cellular oxidative stress increases resulting to an abnormal activation of  glial  .” – please check the definition of oxidative stress. In homeostasis there is no oxidative stress but the redox balance. Secondly it is not only glial aspect but related to all cell types.
  16. Line 230: ROS production is also present in normal conditions in mitochondria and is required for normal functioning. It is a signalling molecule also. Here authors take the process out of proportions. The review should not show only one side, of a strongly pathological aspect. Moreover, ROS does not activate only microglia but reacts with what is the closest. Furthermore, microglia produce ROS as a weapon against the pathogens - they are also the source of ROS.
  17. Line 235: Failure of ATP production and cell death occurs probably in vitro mostly, not in vivo. Even exposure to toxins often does not decrease ATP production in cells due to multiple compensatory mechanisms and safety loops in mitochondria. This would be an extreme situation. Maybe when the process is prolonged in the diseased states the reserves are depleted and cell death could occur.
  18. This chapter on ROS gathers opinions not facts, taken out of proportions. It is not that simple that just antioxidative therapy will fix the cell death. It was documented before that ROS are required to healthy functioning and antioxidant overload was lethal. It is oversimplification generating thought bias that inflammation is just bad and ROS are only bad. Please cover both sides of the story in this review.
  19. Line 251 – Mitophagy is also a natural turnover process – not only the cause of neuroinflammation as you seem to present. Same as fission-fusion.
  20. I don’t understand the figure 1. Mitophagy in resting cells causes their activation? At the same time you write in the description “Mitochondria maintain their healthy and physiological state  by  ” What about neurons? There is no mitophagy in neurons?
  21. Line 288: neither PD or AD does not have “tissue damage” – damage suggest acute, rapid act while neurodegenerative diseases show slowly progressing cell death.
  22. Line 289: “This leads  to  the subsequent  release  of  inflammatory  mediators  and  the  consequent  upregulation  of  the  immune response.  As  described  before,  mitochondria  support  this  unfavorable  ” – this is biased. Mitochondria do not support this  unfavourable  condition. They take part in it. The inflammatory response is not directed to damage neurons but to defend them from a danger. The neuron death is a bystander effect. Like a civilian death in the war between armies - if I may paraphrase. By the way – only neurons die due to the prolonged over-neuroinflammation? What about other cells?
  23. Line 359: “PD is caused by a progressive loss of dopaminergic neurons in neocortex, substantia nigra and brainstem,..” this sentence needs to be rewritten. Dopaminergic cell loss in cortex is not the main thing in PD.
  24. Line 362: “Interestingly, α-Syn is localized at the mitochondrial surface, where influences mitochondrial structure and functions [120–122].” Another sentence creating bias. Synuclein is located predominantly in neuron terminals, but in some small portion it can also be found in mitochondria. Please try to honestly describe the whole picture not just the small pieces that prove your story. Realistic proportions must be kept in scientific approach.
  25. Line 459: “proinflammatory cytokines production, including IL-1β, has been ascribed to fatty acid metabolism..” What does it mean?
  26. Figure 2 – the information provided is just partial. The misspellings appear.
  27. Line 479: “Immediately after  stroke  onset  neurons  fail  to  sustain  cellular  homoeostasis,  resulting  in  a  sequence  of  hurtful  events  strictly connected  to  mitochondria  functions,  including  excitotoxicity,  acidotoxicity,  ionic  imbalance, oxidative  stress,  inflammation  and  apoptosis”. Also line 501: “ischemic  conditions  are  ascribable  to  mitochondria  dysfunction,  as abnormal  Ca 2+   influx,  ROS  production,  mitochondrial  membrane  permeabilization…” Why any of those mitochondrial functions related to inflammation and cell death were mentioned in the general part of the review?
  28. Line 483 “high energy demand accompanied by inadequate energy reserves, make neurons the most OGD sensitive brain residing  cells” Inadequate? If those are inadequate how does brain even function normally?
  29. Table 1 – it should be clearly divided into the experimentally used substances and clinically approved. The experimental list showed here is very poor as compared to what is really currently tested.

Reviewer 2 Report

In this manuscript, Vezzani et al reviewed the role of mitochondria in regulating neuroinflammation, and their role in acute brain damage, including ischemic stroke and epilepsy, but also neurodegenerative disorders including PD and AD.

The manuscript is well organised: it comprises a good introduction, followed by chapters focusing on the role of mitochondria in neuroinflammation, in neurodegeneration, in ischemic stroke and neuroinflammation, and epilepsy. Finally, current therapies are also discussed.

The topics are well presented and clearly discussed, with appropriated citations in the relevant fields.

Figures are clear and adequate.

No major revisions are required.

Minor comments:

Please justify the text alignment of paragraph 3.3.

Please correct line 281, ”NOS “ with “RNS”, also in figure 1.

Moderate grammar corrections are required throughout the manuscript; therefore I would suggest a careful revision of the English grammar. Here I highlight only some of them. Please change the followings:

  • Line 249: “and is a process that aim to” with “a process aimed to“
  • Line 269-270: “that act receptor” with “that act as receptor”
  • Line 278: “as-trocyte” with “astrocytes”
  • 282: “senti-nel” with “sentinel”
  • Other minor grammar checks are required in line 295(promote), 299 (diseases), 303(present), 323 (cause), 348(exacerbate), 392 (demonstrate), 424 (element), 436(parameters), 466(development), 469 (en-zymes), 592 (release, drive)
  • Line 347: “which has been found to be increased” with “which is increased”
  • 389: “even if” with “although”
  • 392 and 404, 423: “mice models” with “mouse models”
  • 431-432: please change the phrase with “Accordingly, reduced levels of autophagic and mitophagic markers, and an impaired energetic metabolism were observed in human samples obtained by AD-affected patients”.
  • 433: “is compensate” with “is compensated”
  • 452: what is (8892349)?
  • 518”NMDA receptors activation” with “the activation of NMDA receptors”
  • 522”rats model with “rat models”

Round 2

Reviewer 1 Report

General Remarks

Both the text style, content and language were improved in the revised version.

Minor remarks:

  1. Abstract – authors could shortly mention in the abstract the main molecular pathways that are discussed in the review. Sentence similar to that in line 140. This will help reader actually get a grip of what is inside the article.
  2. “Accordingly, it is widely recognized that mitochondria exert a pivotal role in the development of neurodegenerative diseases, such as multiple sclerosis, Parkinson’s and Alzheimer’s diseases, since the  neurodegenerative  process  is  highly  based  on  neuroinflammation  and  tissue  Interestingly, it has been suggested that neuroinflammation, and thus mitochondria (dys)function, have a fundamental role also in acute brain damage, such in ischemic stroke and epileptic seizures.”

This part can be combined in one shorter sentence. For example: It is widely recognized that mitochondria exert a pivotal role in the development of neurodegenerative diseases, such as multiple sclerosis, Parkinson’s and Alzheimer’s diseases, as well as in the acute brain damage, such in ischemic stroke and epileptic seizures.

  1. “In  this  review,  we  will  focus  on  the  role  of  mitochondria  in  regulating  neuroinflammation  in neurodegenerative diseases but also in ischemic stroke and epilepsy, giving a hint on the possible therapeutic approaches targeting mitochondrial pathways involved in inflammation.”  Similarly this part can be shortened by cutting the repetition about the diseases. For example like this: In this review, we will focus on the role of mitochondrial molecular signalling in regulating neuroinflammation in the CNS diseases, giving also a hint on the possible therapeutic approaches targeting mitochondrial pathways involved in inflammation.
  2. Introduction: authors use numerous exclamation words such as : important, fundamental, unique, crucial. If everything is of the most importance it is difficult for the reader to create a realistic perspective on the subject.
  3. Throughout the whole text: When staring a new theme, for example moving from oligodendrocytes to astrocytes, and later to the microglia or mitochondria it would be the best to start from the new paragraph. It will ease the comprehension of the text.
  4. Line 44 – about the oligodendrocytes – “being part  of  glial  cells,  oligodendrocytes  do  not  have  a  striking  role  in  the  physiological neuroinflammation, since they mainly provide physical and metabolic support to neurons promoting the myelinating process”. I would suggest caution with this sentence as many scientist could argue against such strong statement. Especially that in the next sentences authors describe their involvement in inflammation. I propose to shorten it. For example: “As far as we are now aware, oligodendrocytes  do  not  have  a  major  role  in  the  physiological neuroinflammation.
  5. Line 72: “On one  side,  it  induces  the  activation  of  repairing  and regenerating  mechanisms  (i.e.  remyelination),  while  on  the  other,  the  uncontrolled  release  of inflammatory mediators as pro-inflammatory cytokines, reactive oxygen species (ROS), and nitric oxide (NO) boost a chronic neuroinflammatory state.” I suggest to add “.. and is potentially dangerous for the neighbouring cells”. Otherwise it means that inflammation boosts inflammation.
  6. Line 75: “The aberrant release of these inflammatory molecules, together with the consequent up-regulation of immune receptors on the other CNS cells, lead to tissue damage and the consequent activation of peripheral B- and T-cell responses due to the lymphatic system drainage [16].” Do you mean that the CNS inflammation causes peripheral inflammation? Please explain.
  7. Line 183: “Interestingly, mitochondria  disruption represents a great source of cytoplasmic dsDNA. Accordingly, mitochondrial damage is not only a cause,  but is also a consequence of  neuroinflammation, resulting in the release of mtDNA in the cytosol  and  also  in  the  extracellular  ” Unnecessary repetition.
  8. Line 355: “… viruses are associated to the development or exacerbation of MS, due to their ability to control the cGAS-STING pathway” instead of ‘control’ I would use word ‘affect’ or ‘influence’ or ‘act via’.
  9. Line 397: “All these  genes  work  to  govern  mitochondrial  functioning,  thus  strengthening  the  evidence  that mitochondrial damage is strongly involved in PD development” Mitochondrial damage?
  10. Line 460: “Interestingly, NLRP3 has been also associated  to  tau  ” This sentence is too vague.
  11. Line 462: “exhibited sustained NLRP3 levels” Something is missing in here. Please rephrase.
  12. Fig 2. “esogenous DNA”
  13. Line 573: “rotenone, a  mitochondrial  complex  I  inhibitor,  allows  the  loss  of  Ψm” ‘Causes’ rather than ‘allows’.
  14. Line 655: ‘close’ rather than “intimate association”
  15. Line 683: “Interestingly, it has been found that  mitophagy  is  highly  active  in  samples  from  hippocampi  and  temporal  lobe  cortices obtained  from  patients  with  refractory  TLE,  but  it  is  unable  to  remove  damaged  mitochondria completely,  thereby  favouring  neuronal  death”. Does it refer to the mitophagy within glia or neurons.
  16. Line 766: “Fingolimod is a sphingosine-1-phosphate-receptor modulator that avoids the recirculation of autoaggressive lymphocytes” and line 783: “the synthetic  compound  naloxone  avoids  microglia activation” The word ‘avoid’ is misused.
  17. Table 1. The column describing ‘clinically approved’ yes/no is potentially misleading. Some drugs such as minocycline are clinically approved, just not for the referred diseases. This should be clearly stated in the table description.
